# Transcriptional Analysis and Identification of a Peptidoglycan Hydrolase (PGH) and a Ribosomal Protein with Antimicrobial Activity Produced by *Lactiplantibacillus paraplantarum*

**DOI:** 10.3390/ijms252312650

**Published:** 2024-11-25

**Authors:** Jessica J. Hurtado-Rios, Ulises Carrasco-Navarro, Julio Cesar Almanza-Pérez, Monica A. Rincón-Guevara, Edith Ponce-Alquicira

**Affiliations:** 1Departamento de Biotecnología, Universidad Autónoma Metropolitana Unidad Iztapalapa, Av. San Rafael Atlixco 186, Col. Vicentina, Mexico City 09340, Mexico; j.j.hurtado@xanum.uam.mx (J.J.H.-R.); ucn@xanum.uam.mx (U.C.-N.); mrincon@izt.uam.mx (M.A.R.-G.); 2Departamento de Ciencias de la Salud, Universidad Autónoma Metropolitana Unidad Iztapalapa, Av. San Rafael Atlixco 186, Col. Vicentina, Mexico City 09340, Mexico; jcap@xanum.uam.mx

**Keywords:** *Lactiplantibacillus paraplantarum*, antimicrobial protein, protein expression, ribosomal protein, peptidoglycan hydrolase, foodborne pathogens

## Abstract

The growing challenge of antibiotic resistance has intensified the search for new antimicrobial agents. Promising alternatives include peptidoglycan hydrolases (PGHs) and certain ribosomal proteins, both of which exhibit antimicrobial activity. This study focuses on a *Lactiplantibacillus paraplantarum* strain, isolated from fermented meat, capable of inhibiting pathogens such as *Listeria innocua*, *Salmonella Typhimurium*, *Escherichia coli*, *Staphylococcus aureus*, and *Weissella viridescens*. The highest growth and antimicrobial activity were observed at a high nitrogen concentration (5.7 g/L). Two antimicrobial proteins were identified: the 50S ribosomal protein L14 (RP uL14) and 6-phospho-*N*-acetylmuramidase (MupG), a PGH. Partial purification and characterization of these proteins were achieved using SDS-PAGE, zymography, and LC-MS/MS. Transcriptional data (RT-qPCR) showed that higher nitrogen concentrations enhanced MupG expression, while increased carbon concentrations boosted RP uL14 expression. These findings highlight the importance of nutritional sources in maximizing the production of novel antimicrobial proteins, offering a potential path to develop effective alternatives against antibiotic-resistant bacteria.

## 1. Introduction

Lactic acid bacteria (LAB) refers to a group of Gram-positive cocci or bacilli that produce lactic acid as the main product derived from carbohydrate fermentation and are generally recognized as safe. These bacteria do not form spores, are motionless, and are catalase negative. The LAB group includes more than 60 genera, including *Lactococcus*, *Leuconostoc*, *Pediococcus*, *Streptococcus*, *Lactiplantibacillus,* among others, and is most reported in food processing [1]. Homofermentative LAB use glucose as a carbon source to produce lactic acid as the main end glycolysis product; heterofermentative LAB produce lactic acid as well as other metabolites such as acetic acid, formic acid, and ethanol under certain conditions [2,3,4]. LAB constitute a bacterial group that is widely distributed in nature and commonly found in several fermented foods, including dairy, meat, fish, cereals, and vegetables, as they improve the flavor and the nutritional value of foods and can reduce harmful microflora, thus increasing shelf life [5]. In addition, some LAB have been recognized as probiotics because of their action to improve consumer health [2,3].

The genus *Lactobacillus* (newly named *Lactiplantibacillus*), part of the LAB group, is one of the most versatile and prominent within this group. These Gram-positive, non-spore-forming bacteria are acid-tolerant and have a low G+C content (33–52%) in their genome. They ferment carbohydrates, producing mainly lactic acid [6,7]. *Lactiplantibacillus paraplantarum,* formerly known as *Lactobacillus paraplantarum*, belongs to the *Lactiplantibacillus plantarum* taxon alongside species like *Lactiplantibacillus pentosus* and *Lacp. plantarum*, which shows phenotypic and genomic heterogeneity [8]. *Lacp. paraplantarum* is notable for its use as a probiotic and starter culture in fermented foods, as well as its ability to produce bacteriocins, contributing to food safety [9]. Similarly to other LAB, this species requires complex nutrients beyond simple energy and carbon sources, including nucleotides, amino acids, and vitamins, for optimal growth and antimicrobial activity [10]. This facilitates its growth in media such as blood agar, Man Rogosa and Sharpe (MRS) broth, Tryptic Soy Broth (TSB), and Casein Glucose Broth (CGB) [11,12].

The ability of LAB to improve food’s shelf life and inhibit certain pathogenic bacteria is mainly due to the synthesis of organic acids, as well as other metabolites and antimicrobial proteins, including bacteriocins, bacteriocin-like inhibitory substances (BLISs), peptidoglycan hydrolases (PGHs), and the recently reported ribosomal proteins (RPs). PGHs are enzymes capable of hydrolyzing the peptidoglycan network present in the bacterial cell wall; they are generally classified into three groups according to their cleavage site: (1) glycosidases that cleave the backbone of glycan, (2) amidases that cleave the side-chain peptide, and (3) peptidases (endo- and carboxy-peptidases). Glycosidases include *N*-acetylmuramidases that hydrolyze the β-1,4 bond between *N*-acetylmuramic acid and *N*-acetylglucosamine, as well as *N*-acetylgluco-amidases that hydrolyze the N-acetylglucosamine residues. Amidases consist of *N*-acetylmuramyl-*L*-alanine amidases that break the amide bond between the *N*-acetylmuramic acid and the *L*-alanine of the peptide side chain. Endopeptidases cleave the bond between two amino acids within the peptide side chain, while carboxypeptidases are able to break the terminal peptide bond of the peptide chain [13,14,15]. Some PGHs are currently used to inhibit the growth of bacteria of public health importance. In particular, lysozyme is one of the most widely used muramidases in the industry as a preservative in meats, sausages, fish, vegetables, fruits, wine, and powdered milk, in addition to applications in the cosmetics and pharmaceutical industries [16,17]. Another example is lysostaphin, a PGH produced by *Staphylococcus simulans* that has been used in food, veterinary, and human medicine due to its bactericidal effect against *Staphylococcus aureus* [18].

On the other hand, RPs have recently been referred to as moonlighting proteins due to their ability to perform more than one biochemical function within the cell or organism. These proteins participate in protein assembly and translation, but they have also shown inhibitory activity against bacteria, viruses, parasites, fungi, and infectious tumor cells. The mechanism associated with their antimicrobial capacity is still unclear; however, researchers have proposed that their antimicrobial activity may be associated with a membrane depolarization mechanism but also with the intracellular increase in reactive oxygen species (ROS) in target cells [19,20]. The very first record of an RP with antimicrobial activity was made by Pütsep et al. [21]; however, during the last decade, the number of reported RPs with antimicrobial activity has increased. Such is the case of the 50S ribosomal protein L36 (RP bL36) secreted by a LAB, *Pediococcus acidilactici* OSU-PECh-3A, and the ribosomal protein S15 (RP uS15), among others [22,23].

Nowadays, the demand for high-quality foods, minimally processed and free of synthetic chemicals, is increasing worldwide. Consequently, the food industry is seeking natural alternatives to prevent microbial contamination and the reduction of food pathogens such as *Listeria monocytogenes*, *Salmonella*, *S. aureus* methicillin-resistant, and *Escherichia coli,* among others. At the same time, the improper and excessive use of antibiotics, which have been used to prevent and treat infections in humans, animals, and plants, has encouraged the prevalence of drug-resistant pathogens, which is considered a global health risk [24]. Some compounds of a protein nature, such as PGHs (lysozyme) and bacteriocins (nisin, enterocin, plantaricin, pediocin, pentocin, pneumocycline, and sakacin) have been proposed as alternative biopreservatives [25]. However, there are still few antimicrobial proteins available for industrial application. Therefore, the objective of this research was to identify the antimicrobial proteins produced by *Lactiplantibacillus paraplantarum* and to evaluate the influence of the culture media composition on the expression of these antimicrobial proteins.

## 2. Results and Discussion

### 2.1. Isolation of LAB from Commercial Salami

One of the key characteristics observed in LAB was a reduction in pH, an effect primarily attributed to the production of metabolites, including lactic acid. For their isolation, MRS agar supplemented with bromocresol green was used. Colonies that exhibited a yellow zone were identified as LAB, as the color change from bromocresol green to yellow indicates acidic conditions [26]. In addition to this color change, the selected colonies exhibited distinctive LAB phenotypic characteristics, such as a circular shape and creamy texture, and were fully isolated on MRS agar for further analysis.

### 2.2. Screening of Antagonistic Activity of Lacp. paraplantarum

Among the isolated colonies, one demonstrated a greater antagonist spectrum (Table 1). This strain exhibited antagonistic activity against both gram-positive (*Listeria innocua* and *Weissella viridescens*) and gram-negative (*Salmonella typhimurium*. and *E. coli*) pathogens. The observed antagonistic activity could be related to the production of various compounds, including organic acids such as lactic acid, acetic acid, and butyric acid, as well as hydrogen peroxide, PGH, bacteriocins, BLIS, or RPs.

These findings are consistent with previous studies, such as that of Palachum et al. [27], who reported that a strain of *Lacp. plantarum* WU-P19 exhibited antimicrobial activity against both gram-positive and gram-negative pathogens. Similarly, more recent research has shown that *Lacp. paraplantarum* 11, another strain within the *Lacp. plantarum* group, exhibits antagonistic activity against a broad spectrum of pathogens, including gram-positive (*Bacillus cereus*, *Bacillus subtilis*, *Listeria monocytogenes*, *L. innocua*, and *S. aureus*) and gram-negative (*E. coli*, *Enterococcus faecalis*, Salmonella enteritidis, *S. typhimurium*, *Shigella dysenteriae*, and *Pseudomonas aeruginosa*) pathogens [28].

### 2.3. Identification of the Selected Strain with Greater Antagonist Spectrum

The identification of the LAB strain through the amplification and sequencing of the 16S rRNA gene showed similarity percentages of 92%, 92%, and 91% with *Lacp. paraplantarum*, *Lacp. plantarum*, and *Lacp. pentosus*, respectively (See Appendix A). Since these values are below 94.5%, according to Yarza et al. [29], it was not possible to accurately define the genus or species, as a similarity greater than 98.5% is required for a clear identification. Therefore, specific identification was performed by amplifying the *recA* gene, as this constitutive gene has been used by researchers to differentiate between *Lacp. plantarum*, *Lacp. paraplantarum*, and *Lacp. pentosus* [30].

The Polymerase Chain Reaction (PCR) used a single reverse primer (pRev) and three species-specific primers (planF, pentF, and paraF) targeting the *recA* gene, generating amplicons of specific sizes: 318 bp for *Lacp. plantarum*, 218 bp for *Lacp. pentosus*, and 107 bp for *Lacp. paraplantarum*. In Figure 1, the genomic Deoxyribonucleic acid (gDNA) of the strain isolated from salami showed a 107 bp amplification band, identifying it as *Lacp. paraplantarum*.

Additionally, the positive control amplified a 318 bp fragment corresponding to *Lacp. plantarum*. The Matrix/Assisted Laser Desorption/Ionization-Time of Flight Mass Spectrometry (MALDI-ToF-MS) method is used as an additional method to corroborate the species identification. In this study, this method allowed for the corroboration of the identity of the strain previously identified molecularly via the *recA* gene. The identification list provided by the MALDI BioTyper 3.0 software (Bruker Daltonics, Bremen, Germany) identified the strain as *Lacp. paraplantarum* with BioTyper log (scores) values of 2.29, 2.22, and 2.20 (See Appendix A). According to the manufacturer’s criteria, a log (score) greater than 2.00 indicates a high-confidence identification at the genus and species level [31].

### 2.4. Impact of High and Low Nitrogen or Carbon Concentration on Lacp. paraplantarum Growth and Antimicrobial Activity

#### 2.4.1. Impact of the Nitrogen Concentration Level

Various reports showed that the medium composition of the culture media can significantly influence the growth and production of antimicrobial peptides of LAB [32]. As for the source of nitrogen, different sources have been used for LAB cultivation, with peptone, yeast extract, and beef extract being the most suitable for normal LAB growth. The modification in these nitrogen sources, either by substitution, combined use, or changes in their concentration within the culture medium, can substantially affect LAB’s growth, metabolism, and antimicrobial activity [33].

In this study, the nitrogen source concentration provided by peptone biotryptase was modified in the CGB medium, while all other medium components remained unchanged. Figure 2A illustrates the growth kinetics of *Lacp. paraplantarum* under high (N2) and low (N1) nitrogen concentrations in the CGB medium after 24 h of incubation. A significant increase in *Lacp. paraplantarum* growth (*p* < 0.05) was observed under the N2 condition, with a higher nitrogen concentration of 5.7 g/L compared to the control CGB medium control, which maintained its nitrogen concentration of 2.7 g/L. Likewise, significantly higher (*p* < 0.05) antimicrobial activity against *L. innocua* was detected in the N2 medium compared to the antimicrobial activity provided by the control CGB medium (Figure 2B).

On the other hand, significantly lower (*p* < 0.05) growth and antimicrobial activity were observed in the N1 medium with a low nitrogen concentration of 1.8 g/L (Figure 2A,B). Comparable results were reported in 2015 and 2021 by Ooi and collaborators [32,34] for *Lacp. plantarum* I-UL4 and RS5, respectively, where an increase in the nitrogen source concentration was directly proportional to the growth and antimicrobial activity of *Lacp. paraplantarum*. Similarly to the obtained results, Parlindungan et al. [33] demonstrated that the combined use of peptone with either beef or yeast extracts increased the growth of *Lacp. plantarum* B21. However, the antimicrobial activity was not enhanced by the combined use of these nitrogen sources; it was maintained with the separate use of peptone and beef extract but was reduced with yeast extract. Nitrogen sources such as yeast extract, as well as peptone, are mainly composed of amino acids, peptides, proteases, carbohydrates, and salts, while peptides or amino acids may function as precursors or inducers in the growth and synthesis of antimicrobial peptides by LAB [35,36].

Table 2 shows the kinetic parameters of *Lacp. paraplantarum* when cultivated in a growth medium containing different concentrations of nitrogen. The highest O.D. max and antimicrobial activity were 1.39- and 927.85-mm per mg of protein, respectively, and were achieved in the condition N2 (5.7 g/L nitrogen) in comparison to the control CGB medium (2.7 g/L nitrogen) and N1 (1.87 g/L nitrogen) that contributed to the lowest O.D. max and antimicrobial activity, being of 1.08 and 401.75 mm per mg of protein, respectively. Additionally, the condition N2 (5.7 g/L nitrogen) produced the highest specific growth rate (µ) of *Lacp. paraplantarum* (0.50 h^−1^), suggesting that this concentration was optimal for the growth of the strain to yield the highest antimicrobial activity. The results obtained in this study showed that the concentration of nitrogen provided by yeast extract and peptone added to the growth media had a significant impact (*p* < 0.05) on the antimicrobial activity.

It is known that nitrogen is an essential element in the biosynthesis of key cellular components, transcriptional regulation, protein synthesis, chemical reactions, and survival in LAB [37,38,39]. Further transcriptional profiling studies are necessary in order to determine the metabolic and signaling pathways related to nitrogen metabolism and the transcription of antimicrobial peptides and their relationship with carbon metabolic pathways, such as PII and PTS [40].

#### 2.4.2. Impact of the Carbon Concentration Level

In addition to nitrogen, carbon is another essential nutrient required by LAB for optimal growth, metabolism, and synthesis of metabolites in culture media. Glucose, a simple sugar, serves as the primary carbon source for LAB and has been shown to enhance the production of antimicrobial peptides due to its simple structure compared to more complex sugars like sucrose [33,41]. This study evaluated the effect of different glucose concentrations and, consequently, the impact of carbon availability on the growth and antimicrobial activity of *Lacp. paraplanatrum* in CGB medium. Figure 3A shows the growth profiles of *Lacp. paraplantarum* in CGB medium under different carbon concentrations.

The strain exhibited the highest growth in the control condition (6.0 g/L carbon), while growth was lower in both C1 (2.7 g/L carbon) and C2 (8.8 g/L carbon), with C2 demonstrating the highest growth among these latter conditions. In contrast, the antimicrobial activity of the strain increased proportionally with the carbon concentration, as illustrated in Figure 3B, with the highest activity observed under the C2 condition. Similar results were reported by Parlindungan et al. [33] for *Lacp. plantarum* B21, where the growth of the strain was not proportional to increasing the concentration of the carbon source (glucose) in the culture medium. However, antimicrobial activity was not affected and increased proportionally with a rise in glucose concentration. The above may be due to osmotic stress, where a sudden osmotic change in the environment causes a water efflux from the cells, loss of turgor pressure, and reduced growth [42].

Similar findings were found by Lim [43], where an increase in glucose source caused a decrease in the growth of *Lacp. plantarum* KC21, while, on this occasion, the activity was also decreased. Nonetheless, high concentrations of carbon sources have been proven to be the best source of antimicrobial peptide production [32,41]. This may be due to the fact that, for each strain, the increase or decrease in the carbon source is affected differently. The results obtained demonstrate that the antimicrobial activity of *Lacp. paraplantarum* can be affected by increasing the amount of the carbon source (glucose) in the CGB medium.

The kinetic analysis (Table 3) showed that even though *Lacp. paraplantarum* obtained a lower O.D. max in the C2 condition (1.15) concerning the CGB control condition; it presented a higher specific growth. However, the highest specific growth rate (0.46 h^−1^) and level of antimicrobial activity (840.89 mm/mg protein) were detected when carbon concentration was increased in the CGB medium.

Finally, comparing all the conditions in which the concentrations of carbon and nitrogen sources were increased or decreased, higher kinetic values were obtained in condition N2, with an O.D. max of 1.39, a specific growth rate of 0.50 h^−1^, and an antimicrobial activity of 927.85 mm/mg protein. The above could be due to nitrogen being used as a source for antimicrobial activity production, and carbon source was utilized to create cellular material and metabolic products and to preserve cellular functions [44].

### 2.5. Identification of Antimicrobial Ribosomal Protein Produced by Lacp. paraplantarum

Once the partial purification of the antimicrobial RP fraction was obtained, a 16% Tris-tricine-PAGE was performed to characterize its protein profile (Figure 4A). Additionally, zymography was used to identify the protein responsible for the antimicrobial activity within this fraction. The zymogram analysis revealed that only one band from the protein profile of the fraction corresponding to the *Lac. paraplantarum* strain exhibited antimicrobial activity against *Micrococcus lysodeikticus* (Figure 4B). This protein presented an apparent molecular weight of 13.4 kDa, determined by both Tris-tricine-PAGE and the zymogram, and was calculated using the Image Lab 6.0.1 software (Bio-Rad, Hercules, CA, USA).

The band corresponding to 13.4 kDa, separated by Tris-tricine-PAGE (Figure 4A), was identified by tandem mass spectrometry (LC-MS/MS) as the 50S ribosomal protein L14 (RPuL14), which showed five unique exclusive peptides, a protein threshold of 99%, and a peptide threshold of 95%. The RPuL14 has a theoretical molecular weight of 13.3 kDa [45], and it is one of the thirty-three proteins that make up the large subunit (LSU) of the bacterial ribosome. Its main function is to form the factor-binding site at the edge of the inter-subunit cleft of the ribosome [46], and it has been described that, in *Escherichia coli*, L14 is “retro-regulated” and is also regulated by the ribosomal protein uS8 [47,48]. Until a few decades ago, RPs were considered to participate only in the assembly and translation of proteins. However, additional functions have been recently discovered, including the inhibition of tumor cells, viruses, bacteria, fungi, and parasites [20].

Regarding LAB that produce RPs with antimicrobial activity, as of the writing of this paper, only one RP, RP bL36, has been identified in the genus *Lactiplantibacillus* [49]. In this study, RP uL14 has been identified as the second RP with antimicrobial activity produced by the genus *Lactiplantibacillus* reported in the literature. Additionally, other LAB, such as *Lactobacillus sakei*, *Lactobacillus* Hma2N, *Lactobacillus salivarius*, and *P. acidilactici,* synthesize the ribosomal proteins bS21, uL1, bL21, and bL36, respectively. These proteins have demonstrated antimicrobial activity against a variety of pathogens, such as *Enterococcus faecalis*, *L. sakei*, *L. innocua*, *Listeria monocytogenes*, *Listeria seeligeri*, *Staphylococcus epidermidis*, *Melissococcus plutonius*, *Streptococcus pyogenes*, *Streptococcus uberis*, *Enterococcus faecium*, and *E. coli* [22,50,51,52].

Similarly, the activity of RPs produced by microorganisms not belonging to LAB has been documented, such as the ribosomal protein 50S L19 (RP bL19) and the ribosomal protein 30S S1 (RP bS1) from *Streptomyces lividans* TK24 and *Pseudomonas aeruginosa*, respectively [53,54]. Although the mechanism of action of RPs with antimicrobial activity is not yet fully understood, several hypotheses have been proposed. One of them, suggested by de Carvalho et al. [50], is that RPs could interfere with the ribosomal assembly of closely related bacteria. Another hypothesis suggests that these proteins can induce the intracellular production of reactive oxygen species (ROS) in bacteria, which would have harmful effects on microbial components such as DNA, Ribonucleic acid (RNA), proteins, and lipids, thereby compromising bacterial cell membrane integrity and leading to cell lysis [20,55].

### 2.6. Identification of Antimicrobial PGH Produced by Lacp. paraplantarum

In order not to claim the antimicrobial activity of PGH fraction by compounds such as lactic acid, the pH of the supernatant was adjusted to 7.0. The protein profile of this fraction was determined using a 10% Tris-glycine-PAGE, which revealed bands with estimated molecular weights ranging from 37 to 250 kDa (Figure 5A). However, only a single band around 68.2 kDa exhibited antimicrobial activity in the zymogram against *M. lysodeikticus* (Figure 5B). The band displaying activity in the zymogram was excised from the Tris-glycine-PAGE and analyzed by LC-MS/MS. Analysis of the (.RAW) files using MaxQuant software (version 2.4.2.0) identified the protein as 6-phospho-*N*-acetylmuramidase, *N*-terminal, named *MupG*.

The primary function of this PGH is to hydrolyze the MurNAc 6-phosphate-GlcNAc linkage in the peptidoglycan (PG) present in the bacterial cell wall, producing MurNAc 6-phosphate and *N*-acetylglucosamine (GlcNAc) [56]. The regulation of cell wall hydrolases can be divided into three main categories: (1) direct activation or inhibition by regulatory proteins or small molecules; (2) spatial regulation through surface proteins or polymers, or by modifying the peptidoglycan substrate; (3) control of PG hydrolase abundance at the transcriptional or post-transcriptional level. Additionally, the mechanisms underlying the synthesis of cell wall hydrolases remain poorly understood [57]. García-Cano et al. [58,59] identified and studied a MupG produced by *P. acidilactici* ATCC 8042, a LAB that inhibits pathogenic and undesirable bacteria in the food industry, such as *Streptococcus pyogenes*, *Listeria monocytogenes*, and *Staphylococcus aureus*. Years later, another strain of *P. acidilactici* OSU-PECh-3A presented in its supernatant fraction a MupG identified with antimicrobial activity against *S. aureus*, indicating its endopeptidase activity [22]. Another MupG that has been extensively studied is lysozyme, an antimicrobial protein widely distributed in many biological tissues, cells, and body fluids, which is also commonly used in the food and pharmaceutical industries [60].

Finally, other PGHs have been produced by LAB with lytic activity, such as the identification of *N*-acetylmuramoyl-*L*-alanine amidase produced by *L. sakei* isolated from salami, which exhibited antimicrobial activity against Gram-positive and Gram-negative spoilage microorganisms [11].

### 2.7. Impact of Carbon and Nitrogen Concentrations in Culture Media on Antimicrobial Protein Expression

The Reverse Transcription Quantitative Polymerase Chain Reaction (RT-qPCR) analysis allowed us to assess the relative expression levels of the *RP uL14* and *MupG* genes in order to determine whether different carbon and nitrogen concentrations in the CGB medium influenced gene expression, using *recA* as the internal housekeeping reference. Gene expression was evaluated at 24 h of *Lacp. paraplantarum* growth, as this time point corresponded with the highest detected antimicrobial activity. All PCR-amplified products displayed a single peak in their dissociation curves, indicating specific amplification (See Appendix A).

As shown in Figure 6A, RP uL14 was significantly higher (*p* < 0.05) in the C2 condition, where the carbon concentration was elevated, showing a three-fold increase compared to the control CGB medium. In contrast, expression was minimal in the C1 condition (low carbon concentration), as well as in both N1 and N2, representing the lower and higher nitrogen concentrations in the CGB medium, respectively. Interestingly, the 24 h growth of *Lacp. paraplantarum* under different growth conditions—specifically, varying carbon and nitrogen concentrations in the culture medium—did not correlate with the expression of RP uL14. In particular, under condition C2, where the highest expression of RP uL14 was observed, the strain’s growth was lower compared to condition N2, which exhibited the lowest expression of RP uL14 (Figure 6A).

This finding contrasts with the traditional view that increased cell growth is typically associated with higher protein production, as microorganisms are generally thought to enhance ribosomal content to support this process. Consequently, it is expected that increased ribosomal content would lead to higher levels of ribosomal RNA (rRNA) and RPs, as these are the primary components of ribosomes [20,61,62]. However, the reduced growth of *Lacp. paraplantarum* under conditions of increased RP uL14 expression suggests a potential lack of immunity to the antimicrobial activity of this RP. In some microorganisms, immunity genes protect against the antimicrobial effects of their own bacteriocins [63], but, in the case of colicins, a type of bacteriocin produced by *E. coli*, the producing cell is often killed [64].

To date, there have been few reports of the effect of variation in carbon and nitrogen concentration in the culture medium on the expression of RPs, especially with antimicrobial activity. Griffioen and collaborators [65] reported that making a nutritional change in the medium of *Saccharomyces cerevisiae* and increasing glucose concentration in the medium resulted in an effect on the RP gene transcription. This coincides with the results obtained in the present study, where the increase in the glucose source had a proportional effect on the expression of the *RP uL14* gene. The results obtained are the first to demonstrate the effect of high carbon concentrations on the expression of RP uL14 with antimicrobial activity.

Figure 6B shows a significant (*p* < 0.05) increase in *MupG* gene expression when the nitrogen concentration in the CGB medium was modified. Specifically, a 15-fold higher expression was observed in the N2 condition (high nitrogen concentration) compared to the CGB control (unmodified nitrogen concentration). In contrast, altering the carbon concentration (either increasing it in C2 or decreasing it in C1) did not result in a significant change in MupG expression. Additionally, the growth of *Lacp. paraplantarum* at 24 h was proportional to the increase or decrease in carbon or nitrogen concentrations.

At the time of writing this paper, there are no reports on the impact of carbon and nitrogen concentrations in culture media on *MupG* (lysozyme) expression with antimicrobial activity in bacteria. Based on our results, it is suggested that the increased expression of *MupG* in the higher nitrogen concentration may be attributed to the increased availability of biotryptase peptone as a nitrogen source. The medium also contained yeast extract, providing additional amino acids and peptides that likely acted as precursors or inducers for growth and antimicrobial protein synthesis in LAB [35,36]. Previous studies have shown that amino acids can enhance the extracellular production of human lysozyme, which supports our findings, where a greater nitrogen source correlated with a higher level of *MupG* expression [66].

## 3. Materials and Methods

### 3.1. Isolation of the Studied Lactic Acid Bacteria

The studied strain was previously isolated from a commercial salami that was provided by an industrial meat processor (Murgati S.A. de C.V., Naucalpan de Juaréz, Mexico). LAB were isolated using serial dilutions plated on MRS agar (Difco, Detroit, MI, USA) containing bromocresol green (0.0025%) as an indicator of acid formation after incubation at 37 °C under anaerobic conditions for 24 h. Bacterial stocks were prepared in CGB (BD Bioxon, Cuautitlán Izcalli, Edo. Mex., Mexico) with 50% glycerol added and preserved at −80 °C in sealed cryovials. Strains were reactivated in the CGB medium for subsequent evaluations.

### 3.2. Antagonistic Activity

The antagonistic activity of the isolated colonies was tested against various food pathogens, and spoilage microorganisms were evaluated using the delayed antagonism method on a solid nutrient medium, specifically, Tryptic Soy Agar (TSA) (Difco, Detroit, MI, USA) containing dipotassium hydrogen phosphate as a buffering agent [67]. The studied strain, *Lacp. paraplantarum*, was inoculated, forming a central line on the TSA plates, and incubated at 29 °C for 24 h. Subsequently, tested pathogens, including *L. innocua* ATCC 33090, *Weissella viridescens* UAM-MG5, *Salmonella thyphimurium* ATCC 14028, *Leuconostoc mesenteroides* FQ, *E. coli* (EPEC) 2348/69, *Staphylococcus aureus* ATCC 6538, *Brochothrix thermosphacta* NCIB-10018, and *Pseudomonas flourescens* B52, were inoculated perpendicularly to the growth of the studied strain, maintaining a distance of approximately 3 mm. The plates were then incubated at 37 °C for an additional 18 h. Antagonistic activity was determined by the presence of a zone of inhibition around the growth of the tested pathogens, indicating the strain’s ability to inhibit the growth of both Gram-negative and Gram-positive microorganisms. The LAB isolated with a greater antagonistic spectrum against foodborne pathogens was selected for further studies.

### 3.3. Bacterial Identification

The identification of the selected strain was determined by using the 16S rRNA gene sequencing, the *recA* gene sequencing, and mass. For 16S rRNA sequencing, DNA was extracted from an 18 h culture pellet using the Wizard Genomic DNA purification kit (Promega, Madison, WI, USA). The integrity of the DNA was verified via electrophoresis on a 0.8% (*w*/*v*) agarose gel. The 16S rRNA gene was amplified by PCR with a Taq DNA polymerase (Thermo Scientific, Waltham, MA, USA) using universal primers 27f (5′-GTT TGA TCM TGG CTC AG-3′) and 1492R (5′-TAC GGY TAC CTT GTT ACG ACT T-3′) [62]. The PCR products were purified and sequenced by Macrogen Inc. (Seoul, Republic of Korea), and the sequences were analyzed using the Basic Local Alignment Search Tool (BLAST) against the National Center for Biotechnology Information (NCBI) RefSeq database.

Furthermore, species differentiation within the *Lacp. plantarum* taxonomic group was achieved by detecting the *recA* gene, using the method described by Torriani et al. [30], with modifications. This methodology utilized endpoint PCR to analyze *recA* gene polymorphism. Three separate PCR reactions were conducted, each using a specific forward primer designed to target polymorphic sites in the *recA* gene sequences of species within the *Lacp. plantarum* group. The PCR forward primers employed were planF (5′-CCG TTT ATG CGG AAC ACC TA-3′) specifically for *Lacp. plantarum*, pentF (5′-CAG TGG CGC GGT TGA TAT C-3′) for *Lacp. pentosus* and paraF (5′-GTC ACA GGC ATT ACG AAA AC-3′) for *Lacp. paraplantarum*. A common reverse primer, pREV (5′-TCG GGA TTA CCA AAC ATC AC-3′), was used in all reactions. The amplification products with sizes of 318 bp, 218 bp, and 107 bp were species-specific for *Lacp. plantarum*, *Lacp. pentosus*, and *Lacp. paraplantarum*, respectively. The PCR reactions were performed using Platinum™ Taq DNA polymerase (Invitrogen, Waltham, MA, USA), and the amplified products were visualized on a 1.2% agarose gel stained with ethidium bromide. A strain of *Lacp. plantarum* was used as a positive control.

In addition, identification was confirmed using MALDI-ToF-MS at the Divisional Mass Spectrometry Laboratory of the Universidad Autónoma Metropolitana. Fresh colonies from MRS agar plates were processed sequentially with deionized water, ethanol, formic acid, and acetonitrile before analysis; the above was used for inactivation, cell disruption, and protein precipitation. After centrifugation, the supernatant was collected and deposited on the steel plate (Bruker Daltonics Inc., Bremen, Germany) with the matrix (α-cyano-4-hydroxycinnamic acid; Sigma-Aldrich, St. Louis, MA, USA), left to dry at ambient temperature, and the plate was introduced into the MALDI–ToF-MS. The identification results were interpreted using the MALDI BioTyper^TM^ 4.0 software (Data base Bruker Daltonics, Bremen, Germany), which analyzed spectra generated by the automated system. The BioTyper log scores, ranging from 0.00 to 3.00, were used to express the similarity between the MALDI-ToF-MS profiles and database entries [68,69].

### 3.4. Variation of the Carbon or Nitrogen Concentrations in the Culture Medium

In this stage, the studied strain was cultivated in a CGB culture medium composed of glucose (10.0 g/L), biotryptase peptone (20.0 g/L), yeast extract (5.0 g/L), ammonium citrate (2.0 g/L), dipotassium hydrogen phosphate (2.0 g/L), magnesium sulfate (0.1 g/L), manganese sulfate (0.05 g/L), and Tween 80 (1.0 g/L), with a final pH = 6.8 [11,70].

The strain was cultivated at two different high and low levels of nitrogen or carbon, which consisted of the variation of the biotriptase peptone concentration (N1 = 11.0 g/L and N2 = 50.0 g/L) or glucose concentration (C1 = 1.8 g/L and C2 = 17.0 g/L) (Table 4), following the reports made by Ooi et al. [32,34] and Escurra et al. [71], which demonstrated the relevance of these nutritional sources in the growth and antimicrobial activity of *Lacp. plantarum*. In addition, the strain was cultivated in the basal CGB medium as a control to evaluate the effect of the carbon and nitrogen concentrations in the culture medium on growth and antimicrobial activity, as well as for the transcriptional analysis of genes associated with the antimicrobial activity of *Lacp. paraplantarum*.

### 3.5. Impact of the High and Low Nitrogen or Carbon Concentrations on Lacp. paraplantarum Growth

Bacterial growth was assessed in the basal and modified CGB over a 24 h period at 29 °C by measuring the absorbance at 600 nm using a Synergy HT spectrophotometer (BioTek, Winooski, VT, USA) in three replications [72]. The obtained data were fitted to the Gompertz model using the NCSS v.97 program [73].

### 3.6. Impact of the Nitrogen or Carbon Concentrations on the Antimicrobial Activity of Lacp. paraplantarum

The strain was cultivated in the basal and modified CGB over a total 24 h period at 29 °C in order to evaluate the antimicrobial activity using an agar well diffusion assay (AWDA) [74]. First, the supernatant (cell-free extract) fraction was obtained by centrifugation of the culture medium at 8000× *g* for 15 min at 4 °C after 6, 9, 12, 15, 18, and 24 h of incubation. The pH of the supernatant was adjusted to 7.0, filtered through a 0.22 μm sterile membrane (Millipore, Bedord, MA, USA), and stored at −20 °C [22]. The protein concentration in each supernatant was measured using the Bradford method [75], with bovine serum albumin (BSA) used to create the standard curve; absorbance was measured at 595 nm using UV/Vis spectrophotometer (Epoch, Biotek, VT, USA).

To assess antimicrobial activity, TSA soft medium (0.8%) was seeded with *L. innocua* ATCC33090 in logarithmic phase (~10^6^–10^7^ CFU/mL) overlaid with a solid TSA medium (1.5%). Wells of 9 mm in diameter were cut into the agar plates, and the supernatants obtained from the N1, N2, C1, C2, and CGB control media, as well as controls, were loaded into the wells. The plates were incubated at 37 °C for 24 h, with 10% (*w*/*v*) nisin as a positive control. Antimicrobial activity was determined by measuring the inhibition zone, and the specific activity was calculated as the difference between the inhibition halo and the well diameter, expressed in mm per mg of protein [76].

### 3.7. Partial Purification and Characterization of Analysis of the Antimicrobial Proteins

In preliminary studies, the presence of two proteins with antimicrobial activity was detected: one exhibited PGH activity, and the other had a low apparent molecular weight (14 kDa). The PGH activity was further characterized using a turbidimetric assay, which confirmed *N*-acetylmuramidase activity [22]; however, these data are not shown. As a result, different strategies were employed to isolate and further analyze both antimicrobial proteins.

#### 3.7.1. Partial Purification and Characterization of the Antimicrobial Ribosomal Protein Impact of the Nitrogen Concentration Level

The partial purification of antimicrobial RP produced by the studied strain was performed using a pH-dependent adsorption–desorption method, as proposed by Yang et al. [77]. The strain was cultured to the stationary phase in a CGB medium, followed by heating to 70 °C for cell and protease inactivation. The pH was adjusted to 5.0 using 10 M NaOH to facilitate protein adsorption, and the cells were collected by centrifugation at 3100× *g* for 20 min. The resulting cell pellet was washed with phosphate buffer (5 mM sodium phosphate, pH 6.5), resuspended in 100 mM NaCl (adjusted to pH 1.0 using 5% H_3_PO_4_), and stirred to promote protein desorption. The supernatant was dialyzed using a membrane (Spectra/Por^®^ 7, USA, MWCO 1 kDa) to remove salt, filtered through a 0.22 μm sterile membrane (Millipore, MA, USA), and then concentrated by lyophilization and freeze-dried at −66 °C (SCIENTZ-10N; Ningbo Scientz Biotechnology, Ningbo, China). Finally, it was dissolved in 100 mM Tris-HCl buffer at pH 8.0 [78].

Tris-tricine-PAGE was employed to determine the molecular size of the RP fraction, the sample was mixed in 2X sample buffer without reducing agent (100 mM Tris-HCl, pH 6.8, 1% (*v*/*v*) SDS, 0.02% (*w*/*v*) Coomassie Brilliant Blue G-250 (CBB, Bio-Rad, Hercules, CA, USA), and 24% glycerol), and electrophoresis conducted using a Mini-Protean Tetra Cell System [79]. The gel was washed, fixed, stained, and destained for visualization. Zymography was utilized to identify the RP responsible for antibacterial activity against *M. lysodeikticus* as the substrate [22].

The presence of antimicrobial activity was detected by a clear zone in the gel. The final gels were scanned using a digital scanner (Gel-Doc XR system, Bio-Rad, Hercules, CA, USA). The documentation results were evaluated through densitometry using the Image Lab software (Version 6.0, Bio-Rad, Hercules, CA, USA). This software facilitated the estimation of molecular weight and the relative density of each band, which were comparatively analyzed against the bands from protein marker (Bio-Rad, broad range, Catalog No. 161-0318, Hercules, CA, USA). The band corresponding to antibacterial activity was excised and subjected to trypsin digestion. The resulting peptides were then purified and identified through tandem mass spectrometry (LC-MS/MS). This analysis was performed using a microcapillary reversed-phase–pressure liquid chromatography coupled with an LTQ-Orbitrap (ThermoElectron) quadrupole ion trap mass spectrometer equipped with a nanospray interface at the Transcripional Proteomics Laboratory, Institut de recherchers cliniques de Montréal (IRCM), Québec, Canada. The protein identification was validated with Scaffold software (version 4.8.9), with criteria for acceptance based on peptide recognition and probability [80].

#### 3.7.2. Partial Purification and Characterization of the Antimicrobial Peptidoglycan Hydrolase

The partial purification of antimicrobial PGH was performed using membrane filtration techniques [52,81]. The pH of the cell-free supernatant was adjusted to 7.0 and then filtered through a sterile 0.22 µm cutoff membrane (Millipore, MA, USA). Subsequently, the supernatant was separated into a high-molecular-weight fraction (>50 kDa) using an Amicon ultrafiltration system with a 50 kDa cutoff (Millipore, Billerica, MA, USA). Tris-glycine polyacrylamide gel electrophoresis (Tris-glycine-PAGE) was then prepared as described by Laemmli [82]. A sample from the ultrafiltration fraction containing 180 µg of protein was mixed with sample buffer, comprising 3.55 mL deionized water, 1.25 mL 0.5 M Tris-HCl pH 6.8, 2.5 mL glycerol, 2.0 mL 10% (*m*/*v*) SDS, and 0.2 mL 0.5% (*w*/*v*) bromophenol blue. Electrophoresis was carried out at 80 V for 30 min, followed by 120 V for 1.5 h, using a Mini-Protean Tetra Cell System (Bio-Rad, Hercules, CA, USA).

A zymogram of Tris-glycine-PAGE was performed according to García-Cano et al. [58], using 10% (*w*/*v*) polyacrylamide resolving gels containing 0.2% (*w*/*v*) autoclaved *M. lysodeikticus* cells as the substrate. The gels were washed three times with deionized water and gently stirred for 30 min, then incubated at 37 °C in a renaturation buffer. The final gels were processed by scanning and capturing digital images as described earlier. After incubation, the gels were stained and destained to visualize the bands. The molecular weight of the band exhibiting antimicrobial activity was determined using Precision Plus Protein™ Dual Color Standards (Bio-Rad, Catalog No. 1610374, Alcobendas, Spain). The molecular weight and relative density of each band were then analyzed relative to the protein marker. Documentation of the gels was performed using the Gel-Doc™ XR system (Bio-Rad, Hercules, CA, USA), and high-resolution digital images were obtained for further analysis. The band corresponding to the antibacterial protein was excised from the Tris-glycine-PAGE. The resulting peptides were fractionated by LC-M/MS electrospray ionization (OrbitrapTher) at the Laboratorio Nacional Plan TECC in CINVESTAV, Irapuato, Mexico. To identify the antimicrobial protein, the mass spectrometry output (.RAW) files were processed using MaxQuant software [83].

### 3.8. Transcriptional Analysis of the Antimicrobial Ribosomal Protein and Peptidoglycan Hydrolase

#### 3.8.1. RNA Extraction and cDNA Synthesis

The expression analysis of RP and PGH was conducted after 24 h of cultivation, the point at which the highest antimicrobial activity was observed. Bacterial cells from different CGB media (N1, N2, C1, C2, and CGB control) were lysed after centrifugation at 8000× *g* for 15 min at 4 °C. The supernatant was discarded, and the bacterial pellet was washed with 0.9% NaCl. The pellet was then frozen with liquid nitrogen and ground into a powder in a sterile mortar. Approximately 100 mg of this powder was transferred to pre-frozen 1.5 mL microtubes. Total RNA was extracted using TRIzol reagent (Invitrogen, Waltham, MA, USA) according to the manufacturer’s instructions. To eliminate any contaminating gDNA, RNA samples were treated with DNase I (Thermo Scientific, Waltham, MA, USA). RNA was quantified using a UV5-Nano spectrometer (Mettler Toledo, Zurich, Switzerland). Complementary DNA (cDNA) was synthesized from the purified RNA using the RevertAid First Strand cDNA Synthesis Kit (Thermo Scientific, Waltham, MA, USA) following the manufacturer’s protocol. The cDNA samples were quantified as described above.

#### 3.8.2. Quantitative Real-Time PCR (qRT-PCR)

The relative expression of RP and PGH in different carbon and nitrogen sources was evaluated using qRT-PCR. Each qRT-PCR reaction included 100 ng of cDNA, 2× Absolute qPCR SYBR^®^ Green Mix (Thermo Scientific, Waltham, MA, USA), and gene-specific forward and reverse primers: recA (forward primer: 5′-GTC ACA GGC ATT ACG AAA AC-3′; reverse primer: 5′-TCG GGA TTA CCA AAC ATC AC-3′), RP (forward primer: 5′-GTG CCC GTG AAA TCC TTA CT-3′; reverse primer: 5′-ACC GGG TGT TGC TTG TTT A-3′), and PGH (forward primer: 5′-GAA GTC TTC TTC AGC CCT TAC C-3′; reverse primer: 5′-CGC GAT AAG TTT GTG GTT GTG-3′). The reactions were carried out on a Gentier 48E real-time PCR system (TIANLONG, Xi’an, China), with all samples analyzed in triplicate, including both calibrator and non-calibrator controls. The qRT-PCR program recommended by the manufacturer was followed. A melting curve analysis was performed for each plate, with fluorescence readings taken every 5 s between 65 and 97 °C to confirm the amplification of a single product. Gene expression levels were analyzed using the 2^−ΔΔCT^ method [84] and normalized to the *recA* housekeeping gene as an internal reference. Data were presented as the mean ± SD of three experimental replicates, with three technical replicates per experimental replicate, and expressed as relative fold change in expression in response to different carbon and nitrogen sources in the culture media.

### 3.9. Statistical Analysis

All determinations were performed in triplicate, and results were presented as the mean ± standard deviation. Significant differences were evaluated using Duncan’s multiple range test (*p* ˂ 0.05). Statistical analyses were performed using the software NCSS v.97 and XLSTAT (Excel statistics, version 2024).

## 4. Conclusions

In summary, the concentration of carbon and nitrogen in the culture medium significantly affects both the growth and antimicrobial activity of *Lacp. paraplantarum*, demonstrating its broad-spectrum action against pathogenic and undesirable bacteria relevant to the food industry. This study identifies the antimicrobial activity of *Lacp. paraplantarum*-related specific proteins: MupG, a PGH, and RP uL14, a ribosomal protein. Notably, this is one of the first studies to report that the expression of these proteins is influenced by the concentrations of carbon and nitrogen in the culture medium, which are specifically provided by glucose and biotryptase peptone sources. Furthermore, the benefits and risks associated with cloned genes in transfected bacteria merit consideration, as gene cloning via transformation enables the production of recombinant proteins for therapies and vaccines, facilitates genetic research, and is cost-effective due to its speed and efficiency. However, this approach may carry risks, including the potential for gene transfer to other bacteria in the environment, which could impact ecosystems and contribute to antibiotic resistance. Additionally, protein overexpression can be toxic to host bacteria, and the use of genetically modified organisms raises biosafety and ethical concerns. These findings suggest that MupG and RP uL14 could be valuable in combating infections caused by pathogenic and antibiotic-resistant bacteria or as potential food preservatives.

## Figures and Tables

**Figure 1 ijms-25-12650-f001:**
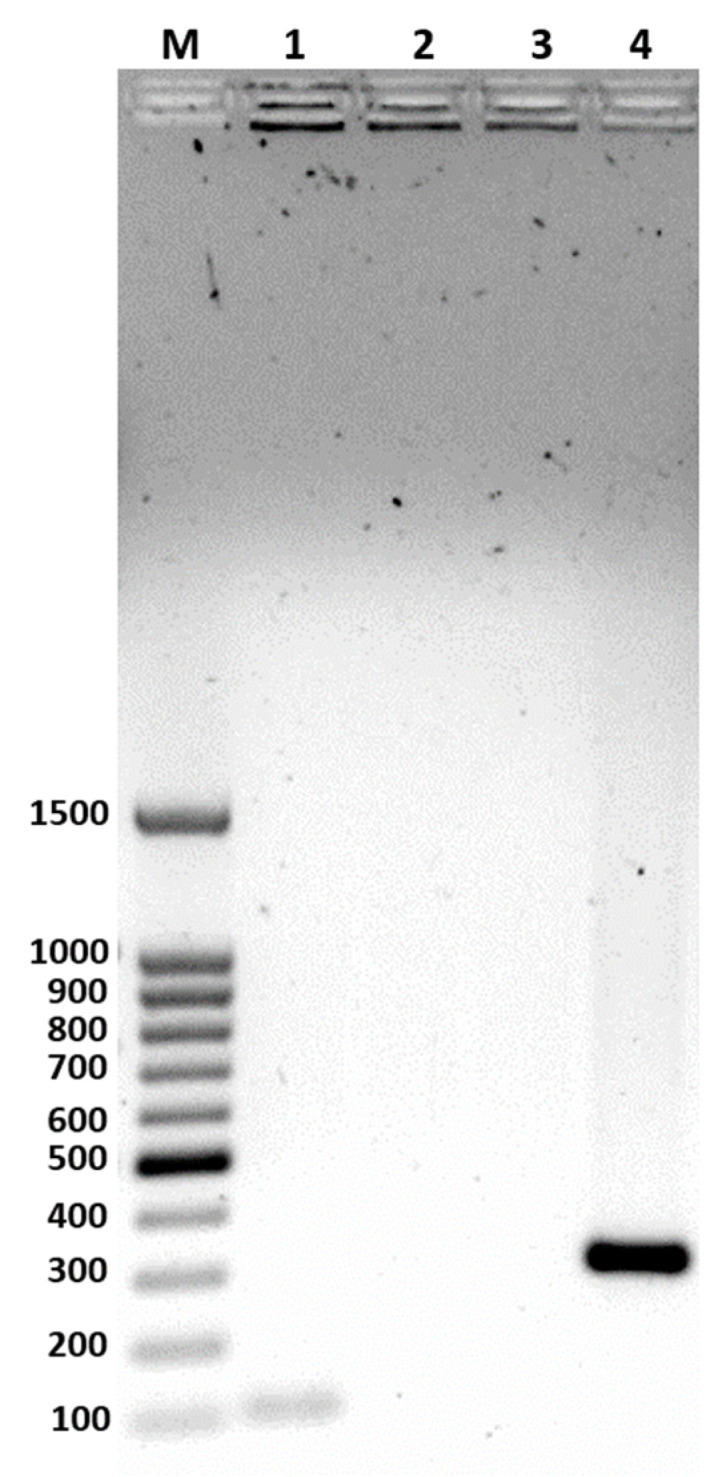
PCR amplification products of the *recA* gene visualized on a 1.2% agarose gel. Lane M: 100 bp DNA ladder (Promega, Madison, WI, USA). Lane 1: amplification with primer paraF (*Lacp. paraplantarum*). Lane 2: amplification with primer planF. Lane 3: amplification with primer pentF. Lane 4: amplification with primer planF (positive control, *Lacp. plantarum*).

**Figure 2 ijms-25-12650-f002:**
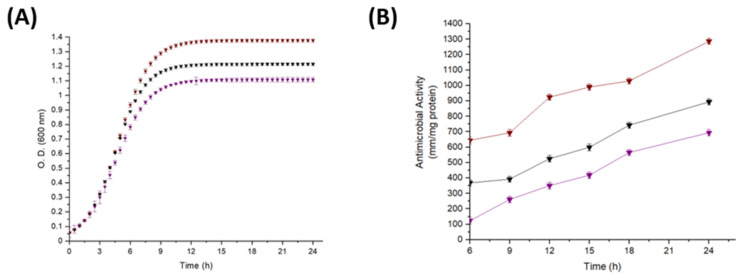
Effect of different nitrogen concentrations in CGB medium on (**A**) growth profile and (**B**) antimicrobial activity of *Lacp. paraplantarum*. CGB control (
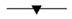
), standard nitrogen concentration (2.7 g/L); N1 (
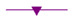
), low nitrogen concentration (1.8 g/L); N2 (
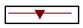
), and high nitrogen concentration (5.7 g/L). Values were presented as mean ± standard error of the mean (SEM), *n* = 3, with vertical bars indicating SEM.

**Figure 3 ijms-25-12650-f003:**
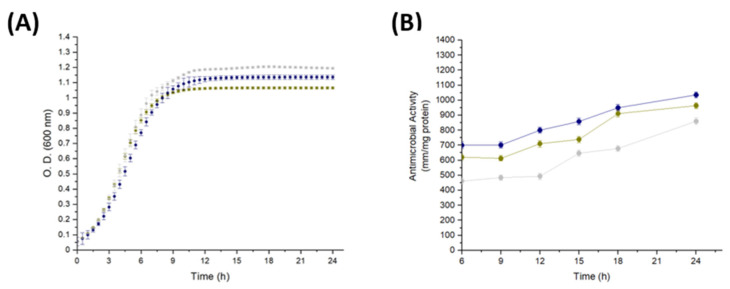
Effect of different carbon concentrations in CGB medium on (**A**) growth profile and (**B**) antimicrobial activity of *Lacp. paraplantarum*. CGB control (
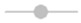
), standard carbon concentration (6.0 g/L); C1 (
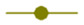
), low carbon concentration (2.7 g/L); C2 (
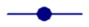
), and high carbon concentration (8.8 g/L). Values were presented as mean ± SEM, *n* = 3, with vertical bars representing SEM.

**Figure 4 ijms-25-12650-f004:**
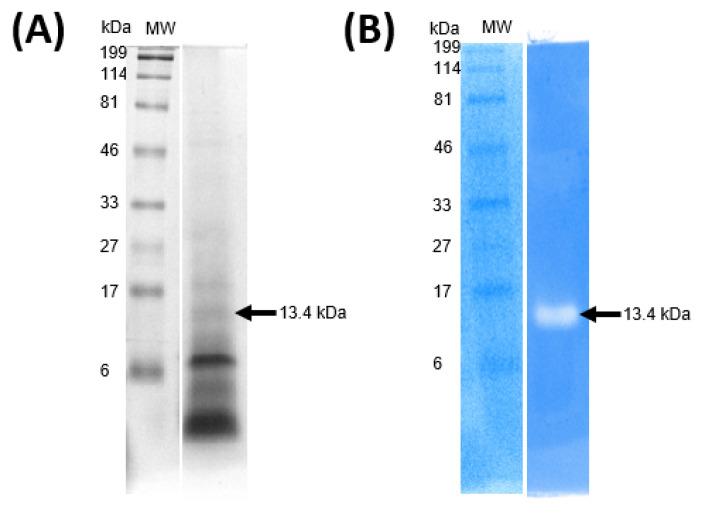
Tris-tricine-PAGE (**A**) and Zymogram (**B**) of antimicrobial ribosomal protein fraction from *Lacp. paraplantarum*. MW, molecular weight marker (Bio-Rad, broad range, Catalog No. 161-0318, Hercules, CA, USA). The band exhibiting antimicrobial activity in the zymogram, analysed through Tris-tricine-PAGE, represents 2.4% of the total densitometry of the bands in the lane by using Image Lab 6.0.1 software (Bio-Rad, Hercules, CA, USA).

**Figure 5 ijms-25-12650-f005:**
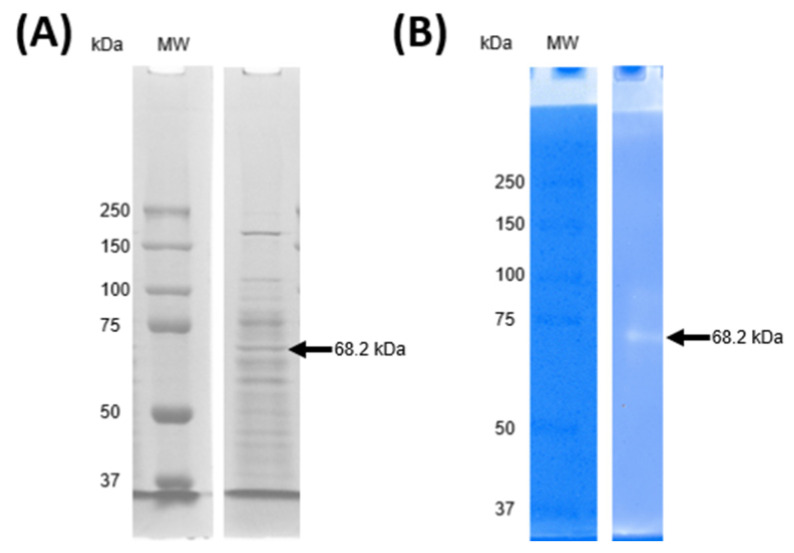
Tris-glycine-PAGE (**A**) and Zymogram (**B**) of antimicrobial PGH fraction from *Lacp. paraplantarum*. MW, molecular weight marker Precision Plus Protein™ Dual Color Standards (Bio-Rad, Catalog No. 1610374, Alcobendas, Spain). The band exhibiting antimicrobial activity in the zymogram, analyzed through Tris-glycine-PAGE, represents 6.7% of the total densitometry of the bands in the lane by using Image Lab 6.0.1 software (Bio-Rad, Hercules, CA, USA).

**Figure 6 ijms-25-12650-f006:**
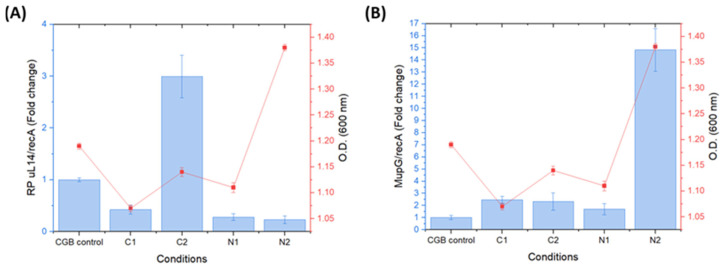
Relative expression levels of the *RP uL14* (**A**) and *MupG* (**B**) genes in *Lacp. paraplantarum* grown in CGB medium with different carbon and nitrogen concentrations. The control condition (CGB) contains 6.0 g/L carbon and 2.7 g/L nitrogen. C1, low carbon (2.7 g/L). C2, high carbon (8.8 g/L). N1, low nitrogen (1.8 g/L). N2, high nitrogen (5.7 g/L). Gene expression was normalized to *recA* gene using the 2^−ΔΔCt^ method. The red line graph shows cell growth (O.D. 600 nm) at 24 h. Data are presented as mean ± SEM (*n* = 3), with vertical bars indicating SEM.

**Table 1 ijms-25-12650-t001:** Antibacterial activity of the isolated LAB.

Bacteria	Gram	Antagonistic Activity ^1^
*Listeria innocua* ATCC 33090	Positive	+
*Weissella viridescens* UAM-MG5	Positive	+
*Salmonella typhimurium* ATCC 14028	Negative	+
*Leuconostoc mesenteroides* FQ	Positive	-
*Escherichia coli* (EPEC) 2348/69	Negative	+
*Staphylococcus aureus* ATCC 6538	Positive	-
*Brochothrix thermosphacta* NCBI-10018	Positive	-
*Pseudomonas flourescens* B52	Negative	-

^1^ (+) positive antagonistic activity with the presence of a zone of inhibition (-) negative antagonistic activity.

**Table 2 ijms-25-12650-t002:** Duncan test comparisons of kinetic parameters and antimicrobial activity at different nitrogen (N) concentrations in CGB medium.

Parameter	Condition		
	N1 (1.87 g/L N)	CGB Control (2.7 g/L N)	N2 (5.7 g/L N)
O.D. max (ʎ = 600 nm)	1.08 ± 0.08 ^a^	1.20 ± 0.03 ^b^	1.40 ± 0.07 ^c^
μ (h^−1^)	0.48 ± 0.01 ^a^	0.45 ± 0.01 ^b^	0.50 ± 0.01 ^c^
Antimicrobial Activity (mm/mg protein)	401.75 ± 34.99 ^a^	586.57 ± 41.24 ^b^	927.85 ± 66.90 ^c^

Values are represented as means ± standard deviations from three independent experiments. Duncan’s multiple range test was used for comparison, with significance at *p* < 0.05. Different superscript letters denote significant differences within the same row.

**Table 3 ijms-25-12650-t003:** Duncan test comparisons of kinetic parameters and antimicrobial activity at different carbon (C) concentrations in the CGB medium.

Parameter	Condition		
	C1 (2.7 g/L C)	CGB Control (6.0 g/L C)	C2 (8.8 g/L C)
O.D. max (ʎ = 600 nm)	1.07 ± 0.02 ^a^	1.20 ± 0.03 ^b^	1.15 ± 0.03 ^c^
μ (h^−1^)	0.43 ± 0.01 ^a^	0.44 ± 0.01 ^b^	0.46 ± 0.01 ^c^
Antimicrobial Activity (mm/mg protein)	759.64 ± 51.92 ^a^	603.89 ± 33.34 ^b^	840.89 ± 40.62 ^c^

Values are represented as means ± standard deviations from three independent experiments. Duncan’s multiple range test was used for comparison, with significance at *p* < 0.05. Different superscript letters denote significant differences within the same row.

**Table 4 ijms-25-12650-t004:** Composition of Casein Glucose Broth (CGB) medium at different concentrations of nitrogen and carbon sources.

Composition		Concentration (g/L)			
	C1	C2	Control	N1	N2
Peptone biotryptase	20	20	20	11	50
Glucose	1.8	17	10	10	10

## Data Availability

The data are not publicly available due to privacy regulations designed to protect the privacy and consent of research participants. However, the data supporting the findings of this study can be made available upon reasonable request to the corresponding authors.

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
