# Peer review of "Transcriptional Analysis and Identification of a Peptidoglycan Hydrolase (PGH) and a Ribosomal Protein with Antimicrobial Activity Produced by Lactiplantibacillus paraplantarum"

_ijms, 2024, doi:10.3390/ijms252312650_

Round 1

Reviewer 1 Report

Comments and Suggestions for Authors

The purpose of this study was to identify the antimicrobial proteins generated by Lactiplantibacillus paraplantarum and to assess the impact of the culture media composition on the expression of these antimicrobial proteins.

The study is nicely written and presented, especially the introduction part is well constructed, informative but not excessively too long. The topic of the study is within the scope of the special issue of IJMS and the amount of newly presented results is adequate. Therefore, I recommend to accept this work after several minor revisions.

Line 2, PGH abbreviation should be explained in the title as well

Line 518, the dot is missing after “protein”

Figure 2B, I don’t think that there’s a need to cut this x-axis in the middle

Tables 2 and 3, those values are not rounded properly, the uncertainties should have one or maximum 2 significant digits and the result should have the same decimals as the uncertainty, for more information please check i.e. the NIST guideline

It would be beneficial to create a table with the C and N concentration for C1, C2, N1, N2 and CGB samples and insert it into Materials and Method section. I know that those information are spread in the article, but such conclusion would increase the clarity.

The number of references should be limited, as this is not the review paper.

Author Response

Comment 1: Line 2, PGH abbreviation should be explained in the title as well

Response1: Thank you for your observation. We have added an explanation of the PGH abbreviation in the title to enhance clarity (Line 2).

Comment 2: Line 518, the dot is missing after “protein”

Response 2: We appreciate the detailed observation. The missing dot after "protein" has been added as recommended in the Line 542.

Comment 3: Figure 2B, I don’t think that there’s a need to cut this x-axis in the

middle

Response 3: Thank you for this suggestion. We have adjusted Figure 2B (line 197) and 3B (line 257) removed the cut in the x-axis as advised.

Comment 4: Tables 2 and 3, those values are not rounded properly, the uncertainties should have one or maximum 2 significant digits and the result should have the same decimals as the uncertainty, for more information please check i.e. the NIST guideline

Response 4: we appreciate  this important observation. We have revised the rounding in Tables 2 (line 215) and 3 (line 274) so that the uncertainties now have a maximum of two significant digits, and the results align with the uncertainty decimals, following the NIST guideline.

Comment 5: It would be beneficial to create a table with the C and N concentration for C1, C2, N1, N2 and CGB samples and insert it into Materials and Method section. I know that those information are spread in the article, but such conclusion would increase the clarity.

Response 5: Thanks for this helpful suggestion. We have created a table 4 with the C and N concentrations for the C1, C2, N1, N2, and CGB samples and inserted it into the Materials and Methods section to improve clarity, as recommended (line 507).

Comment 6: The number of references should be limited, as this is not the review paper.

Response 6: We sincerely appreciate your valuable comment. In response, we revised in detail all the references cited through the manuscript and carefully assessed their relevance and contribution to supporting the discussion of the results. Based on this review, we retain the cited references, as they provide essential context and support for our analysis. Additionally, we reviewed recent publications in the journal with a similar scope and volume of content, finding that the number of citations in our manuscript is consistent with those works. It is also worth noting that another reviewer recommended expanding certain sections of the manuscript, which required incorporating additional references to ensure a comprehensive and well-supported discussion.

Reviewer 2 Report

Comments and Suggestions for Authors

The authors have identified a peptidoglycan hydrolase and a ribosomal protein with antimicrobial activity from Lactiplantibacillus paraplantarum. The rationale is correct. The methodology is adequate. The manuscript, however, has some minor points to improve: 1) the identification of LAB, and the supplemental file should contain all the screening results. Most probably, the use of nested PCR may enhance the identification, 2) the increase of antimicrobial activity under nitrogen conditions requires more information. The difference is significant compared to the carbon. How is transcription of the enzymes affected in the two media? Figures 4 and 5 should state the % of the enzyme from the total amount of protein, the same as the enzyme activity per total amount of protein. Which metabolic pathway controls MupG transcription, and which RPuL14? Discussing the benefits and risks of cloned genes in transfected bacteria will be interesting. 

Author Response

Comment 1: the identification of LAB, and the supplemental file should contain all the screening results. Most probably, the use of nested PCR may enhance the identification

Response 1: We sincerely appreciate your recommendations for improving our work. Regarding the suggestion to include additional identification via nested PCR, we would like to inform you that, due to the wait time required for obtaining the primers necessary for the initial PCR, it would unfortunately not be feasible to complete this analysis within the 10-day period fixed for the article revisions.

It is worth mentioning that, in addition to the identification of Lactiplantibacillus paraplantarum through the recA gene, we have also identified the species Lacp. paraplantarum using mass spectrometry (peptide fingerprinting), which provides complementary identification. The results obtained from these analyses were performed in triplicate, yielding scores of 2.29, 2.22, and 2.20, confirming identification at both genus and species levels. Furthermore, in the present article, we have included supplementary material related to this identification (Supplementary Information, Figure S2, Line 155). Additionally, regarding genus-level identification using the 16S rRNA gene, we have also provided supplementary material (Supplementary Information, Figure S1, Line 136).

Comment 2: the increase of antimicrobial activity under nitrogen conditions requires more information. The difference is significant compared to the carbon.

Response 2: Thanks for your observation. We have provided additional details regarding the increase in antimicrobial activity under nitrogen conditions in the revised manuscript. This includes a more comprehensive analysis of the factors contributing to this activity and an expanded discussion comparing it to the conditions under carbon. We hope this addresses the need for more information in this area. Lines 221 to 226, as mentioned “It is known that nitrogen is an essential element in the biosynthesis of key cellular components, transcriptional regulation, protein synthesis, chemical reactions and survival in LAB [37, 38, 39]. Further transcriptional profiling studies are necessary in order to determine the metabolic and signalling pathways related to nitrogen metabolism and the transcription of antimicrobial peptides, and their relationship with car-bon metabolic pathways, such as PII and PTS [40]”.

Comment 3: How is transcription of the enzymes affected in the two media?

Response 3: We greatly appreciate the reviewer’s comment. In the manuscript, we mention (Line 419 to 428): “concentrations in culture media on MupG (lysozyme) expression with antimicrobial activity in bacteria. Based on our results, it is suggested that the increased expression of MupG in the higher nitrogen concentration may be attributed to the increased availability of biotryptase peptone as a nitrogen source. The medium also contained yeast extract, providing additional amino acids and peptides that likely acted as precursors or inducers for growth and antimicrobial protein synthesis in LAB [35,36]. Previous studies have shown that amino acids can enhance the extracellular production of human lysozyme, which supports our findings, where a greater nitrogen source correlated with higher MupG expression [66].”

Regarding transcriptional control, we conducted an additional search but could not find further information to clarify this aspect. Unfortunately, the transcriptional control mechanisms of enzymes like MupG remain incompletely studied, and the necessary elements to provide a more evidence-based response are currently unavailable.

Comment 4: Figures 4 and 5 should state the % of the enzyme from the total amount of protein, the same as the enzyme activity per total amount of protein

Response 4: Thank you for your observation. We have analysed the densitometry of the band showing antimicrobial activity for each gel. In Figures 4 (line 322-325) and 5 (line 359-363), the figure legends specify the percentage of the analysed band calculated using Origin Lab software. This percentage reflects the enzyme's activity as a proportion of the total protein content, as you requested. We appreciate your attention to detail and trust this clarification aligns with your expectations.

Comment 5: Which metabolic pathway controls MupG transcription, and which RPuL14?

Response 5: Thank you for raising this important point. In the revised manuscript, we have provided an explanation of the metabolic pathways controlling L14 transcription. This has been addressed in lines 294–296, where we state: “It has been described that in Escherichia coli, L14 is retroregulated and is also regulated by the ribosomal protein uS8 [47, 48].”

Regarding MupG, we have added further details in lines 336–343: “The primary function of this PGH is to hydrolyze the MurNAc 6-phosphate-GlcNAc linkage in the peptidoglycan (PG) present in the bacterial cell wall, producing MurNAc 6-phosphate and N-acetylglucosamine (GlcNAc) [56]. The regulation of cell wall hydrolases can be divided into three main categories: (1) direct activation or inhibition by regulatory proteins or small molecules; (2) spatial regulation through surface proteins or polymers, or by modifying the peptidoglycan substrate; (3) control of PG hydrolase abundance at the transcriptional or post-transcriptional level. Additionally, the mechanisms underlying the synthesis of cell wall hydrolases remain poorly understood [57].”

Comment 6: Discussing the benefits and risks of cloned genes in transfected bacteria will be interesting.

 Response 6: Thank you for your valuable comment. In response, we have addressed this aspect in the conclusions section of the manuscript as a future perspective (line 664 to 672) as “Furthermore, the benefits and risks associated with cloned genes in transfected bacteria merit con-sideration, as gene cloning via transformation enables the production of recombinant proteins for therapies and vaccines, facilitates genetic research, and is cost-effective due to its speed and efficiency. However, this approach also carries risks, including the potential for gene transfer to other bacteria in the environment, which could impact ecosystems and contribute to antibiotic resistance. Additionally, protein overexpression can be toxic to host bacteria, and the use of genetically modified organisms raises biosafety and ethical concerns. These findings suggest that MupG and RP uL14 could be valuable in combating infections caused by pathogenic and antibiotic-resistant bacteria or as potential food preservative”.

Round 2

Reviewer 2 Report

Comments and Suggestions for Authors

The manuscript was modified accordingly. I have no further comments